# Post-transcriptional regulation of aromatic amino acid metabolism by GcvB small RNA in *Escherichia coli*

Takeshi Kanda,[1,2] Toshiko Sekijima,[3] Masatoshi Miyakoshi[1,2,3]

**ABSTRACT** *Escherichia coli* synthesizes aromatic amino acids (AAAs) through the common pathway to produce the precursor, chorismate, and the three terminal pathways to convert chorismate into Phe, Tyr, and Trp. *E. coli* also imports exogenous AAAs through five transporters. GcvB small RNA post-transcriptionally regulates more than 50 genes involved in amino acid uptake and biosynthesis in *E. coli*, but the full extent of GcvB regulon is still underestimated. This study examined all genes involved in AAA biosynthesis and transport using translation reporter assay and qRT-PCR analysis. In addition to previously verified targets, *aroC*, *aroP*, and *trpE*, we identified new target genes that were significantly repressed by GcvB primarily via the R1 seed region. Exceptionally, GcvB strongly inhibits the expression of *aroG*, which encodes the major isozyme of the first reaction in the common pathway, through direct base pairing between the *aroG* translation initiation region and the GcvB R3 seed sequence. RNase E mediates the degradation of target mRNAs except *aroC* and *aroP* via its C-terminal domain. GcvB overexpression prolongs the lag phase and reduces the growth rate in minimal media supplemented with AAAs and confers resistance to an antibiotic compound, azaserine, by repressing AAA transporters.

**IMPORTANCE** *E. coli* strains have been genetically modified in relevant transcription factors and biosynthetic enzymes for industrial use in the fermentative production of aromatic amino acids (AAAs) and their derivative compounds. This study focuses on GcvB small RNA, a global regulator of amino acid metabolism in *E. coli*, and identifies new GcvB targets involved in AAA biosynthesis and uptake. GcvB represses the expression of the first and last enzymes of the common pathway and the first enzymes of Trp and Phe terminal pathways. GcvB also limits import of AAAs. This paper documents the impact of RNA-mediated regulation on AAA metabolism in *E. coli*.

**KEYWORDS** small RNA, GcvB, amino acid biosynthesis, amino acid transport

Aromatic amino acids (AAAs), namely L-phenylalanine (Phe), L-tyrosine (Tyr), and L-tryptophan (Trp), are important for food, pharmaceutical, and chemical industries. *Escherichia coli* is genetically engineered in its biosynthesis and transport pathways and is widely used for fermentative production of AAAs (1–4). The biosynthetic pathway of AAAs is composed of the common pathway and three terminal pathways (5). The common pathway, a.k.a. the shikimate pathway, starts from the condensation of phosphoenolpyruvate and erythrose 4-phosphate, which is catalyzed by 3-deoxy-D-arabino-heptulosonate 7-phosphate (DAHP) synthase isozymes, AroG, AroF, and AroH (Fig. 1). DAHP is then converted to chorismate, the common precursor of the three AAAs, through six enzymatic reactions. Chorismate is transformed to phenylpyruvate and 4-hydroxyphenylpyruvate by chorismate mutase-prephenate dehydratase (PheA) and chorismate mutase-prephenate dehydrogenase (TyrA) and subsequently to Phe and Tyr by aromatic aminotransferase (TyrB), respectively. Trp is produced from chorismate via

**Peer Reviewer** Tanmay Dutta, Indian Institute of Technology Delhi, New Delhi, Delhi, India

Address correspondence to Masatoshi Miyakoshi, mmiyakoshi@md.tsukuba.ac.jp.

The authors declare no conflict of interest.

See the funding table on p. 16.

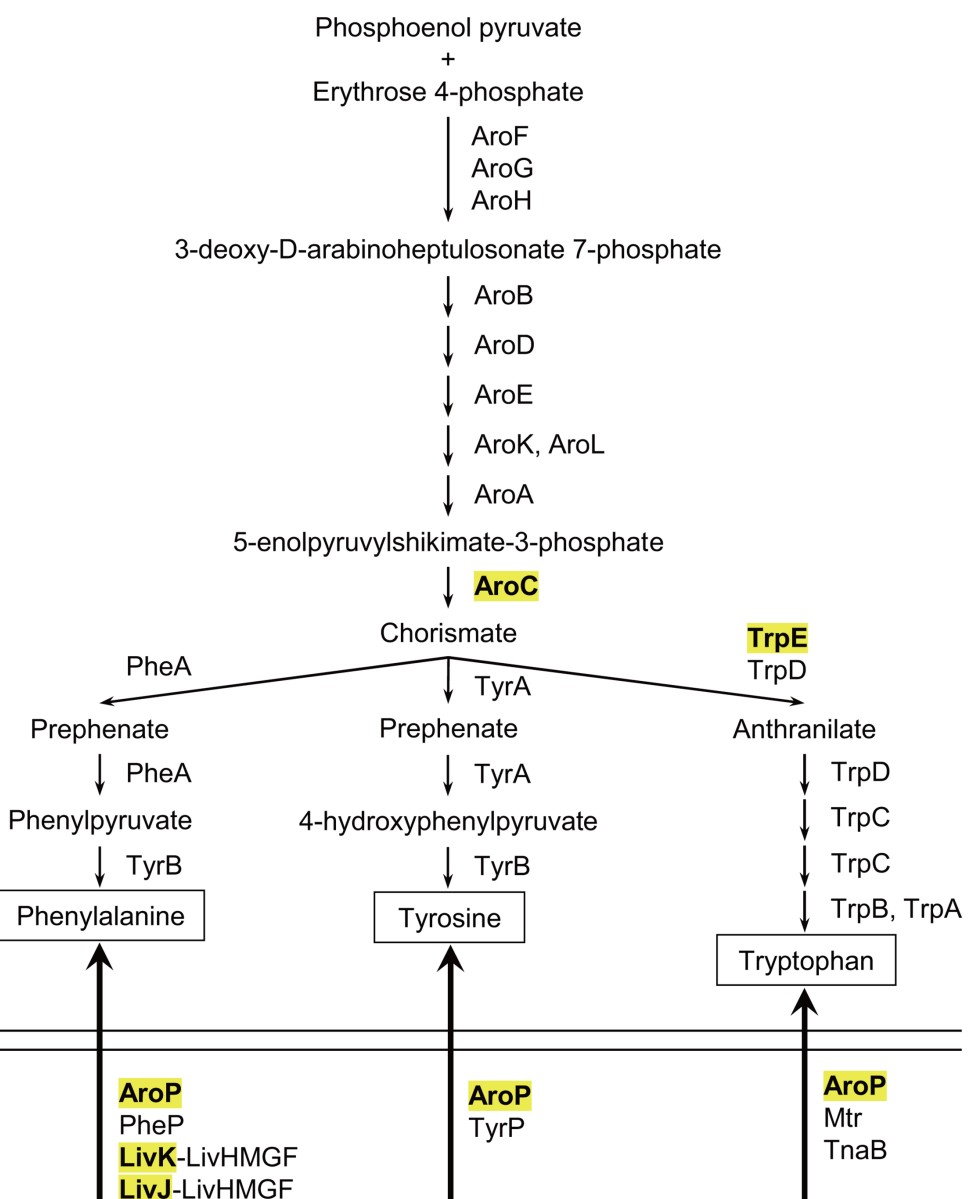

**FIG 1** Schematic representation of AAA biosynthesis and import pathway in *E. coli*. The proteins whose expression is directly regulated by GcvB are highlighted in yellow.

anthranilate by the enzymes encoded in the *trpEDCBA* operon. Since the biosynthesis of AAAs requires chemical intermediates from glycolytic pathway and thus is costly (6), *E. coli* imports exogenous AAAs through five different transport systems when available (Fig. 1). The general permease AroP transports all three AAAs with high affinities, but *E. coli* also possesses four dedicated transporters: PheP for Phe, TyrP for Tyr, and Mtr and TnaB for Trp (5).

Fundamental study of the regulatory mechanism underlying the metabolic pathways is crucial to improving *E. coli*'s efficiency and yields for the fermentative production of valuable compounds. The main control of AAA biosynthetic pathways is feedback inhibition of the first reaction in each pathway by the end products (5). Three isozymes in the common pathway, AroG, AroF, and AroH, are allosterically inhibited by Phe, Tyr, and Trp, respectively. PheA, TyrA, and TrpE of the branched pathways are also allosterically inhibited by the respective end products. Next, the gene expression of AAA biosynthetic enzymes and transporters is regulated mainly at the level of transcription.

Two transcriptional regulatory proteins, TyrR and TrpR, control the transcription of genes dispersed at different loci in the *E. coli* genome (5, 7). In addition, the *pheA* and *trpE* genes are preceded by transcriptional attenuators, in which the leader mRNAs contain consecutive Phe and Trp codons (8).

Transcriptional and post-transcriptional regulation act together to fine-tune the expression of amino acid metabolic genes (9). The global regulatory small RNA (sRNA) GcvB is transcriptionally regulated by the glycine cleavage pathway regulators, GcvA and GcvR (10), and is one of the most abundantly expressed sRNAs bound with the RNA chaperone Hfq in nutrient-rich media. GcvB contains three seed sequences, R1, R2, and R3, to hybridize with its target mRNAs *in trans* and represses their expression mainly at the level of translation initiation (11–19). By compiling the global RNA–RNA interactome data sets (20–23), we have estimated that GcvB interacts with >50 mRNAs in *E. coli*, the majority of which are associated with transport and biosynthesis of many but not all amino acids (24). However, amino acid biosynthetic genes are underrepresented in the data sets due to their low expression levels during growth in nutrient-rich media.

Regarding AAA biosynthesis and transport, we have verified that GcvB inhibits the expression of *aroC*, *aroP*, and *trpE* at the post-transcriptional level (24). However, it is possible that GcvB redundantly regulates more genes in the same pathway. This study aimed to determine the full extent of GcvB targets involved in the AAA metabolism in *E. coli*. To this end, we constructed translational reporters for all AAA biosynthetic and transporter genes and validated additional genes that are post-transcriptionally inhibited by GcvB via either R1 or R3 seed sequence.

## RESULTS

### Regulation of the common biosynthetic pathway

We have previously shown that GcvB post-transcriptionally represses the expression of AroC (24), which catalyzes the last step in the common biosynthetic pathway to produce chorismate, the precursor of AAAs. To investigate the post-transcriptional regulation of the other enzymes in the common pathway, we applied the established translational GFP reporter system (25, 26). The translational fusions with the superfolder GFP (sfGFP) were constructed based on pXG-10sf and pXG-30sf plasmids (Tables S1 through S4). Here, we used *E. coli* BW25113 Δ*gcvB*Δ*sroC* mutant as the host strain for the reporter plasmids and the compatible GcvB-expressing plasmids because the SroC sRNA induces the RNase E-mediated degradation of GcvB by binding at two distant regions (27) and may alter the steady-state levels of GcvB variants in this assay (24).

In *E. coli*, the first reaction is catalyzed by the three DAHP synthase isozymes, AroG, AroF, and AroH, among which AroG contributes approximately 80% of DAHP production (28). We found that the expression level of *aroG* was reduced to ~20% upon GcvB expression, while *aroF* and *aroH* were not significantly regulated (Fig. 2A). The translation initiation region of *aroG* was predicted to base pair with the R3 seed region of GcvB by the IntaRNA program (29). To verify the interaction between *aroG* and R3, we took advantage of the previously constructed plasmids expressing various GcvB mutants (24). The repression of *aroG* was slightly relieved by deletion of the major R1 seed region (ΔR1) and was fully abrogated by further deletion of R3 (ΔR1ΔR3) (Fig. 2A). Moreover, a five-nucleotide replacement (mutR3) and single-nucleotide substitutions (G156C, G160C, and C162G) in the R3 seed sequence disrupted the repression by GcvBΔR1 as expected. Finally, the G-8C mutation in the Shine–Dalgarno sequence of *aroG* ($aroG_{G-8C}$), which reduced the translation efficiency by ~10-fold, disabled the repression by GcvBΔR1, but the complementary mutation C162G in GcvBΔR1 partially restored the repression of $aroG_{G-8C}$ (Fig. 2A). These results indicate that the R3 seed region of GcvB directly interacts with the translation initiation region of *aroG* to inhibit its expression.

The remaining six genes of the common pathway were also examined by the two-plasmid reporter assay. GcvB moderately repressed *aroB* (64%) and *aroE* (51%), but *aroD*, *aroK*, and *aroA* were not significantly affected (Fig. 2B). Deletion of R1 slightly relieved the repression of *aroB* and *aroE*, but we could not find firm base-pairing

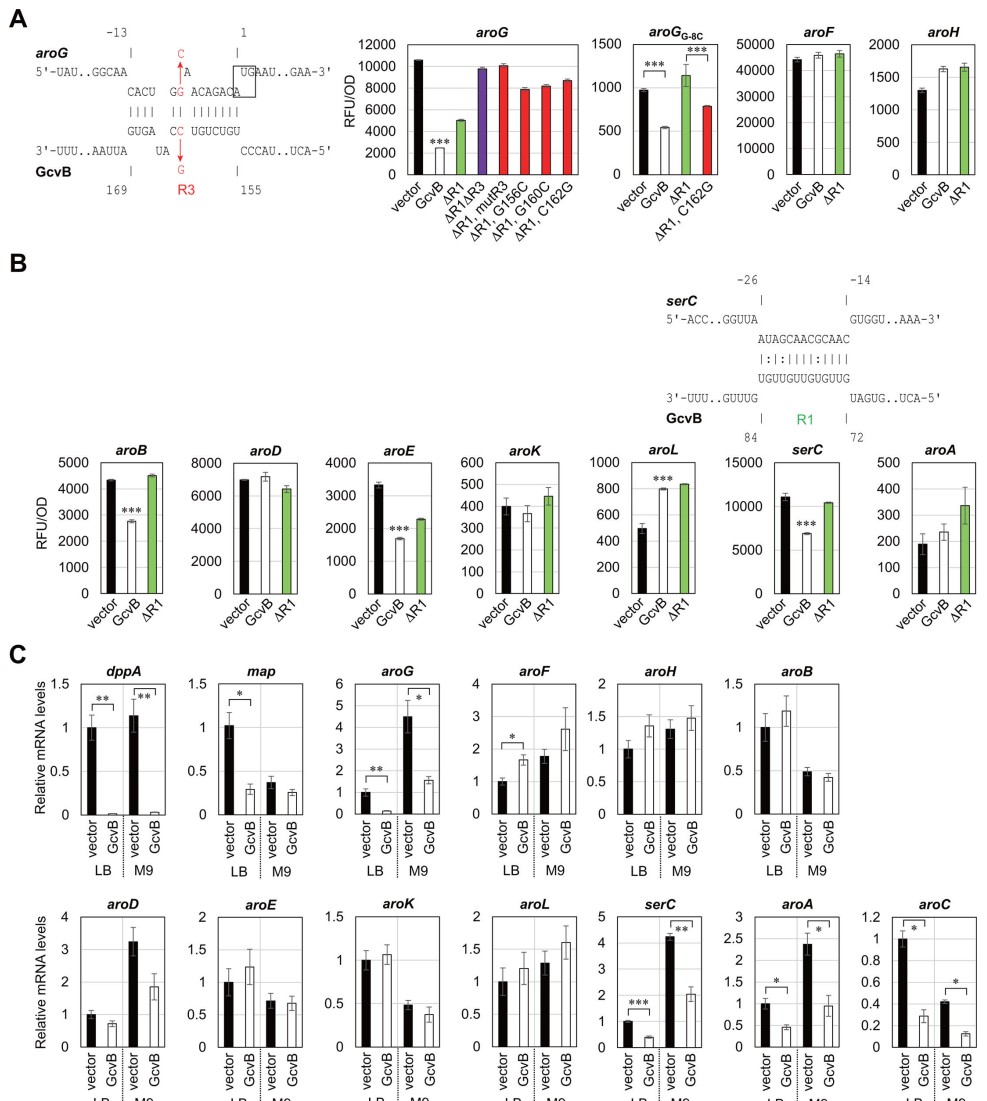

**FIG 2** Post-transcriptional regulation of the common biosynthetic pathway mediated by GcvB. (A, B) GFP reporter assays in *E. coli* Δ*gcvB*Δ*sroC* strain harboring pTP11 (vector), pP_L-*gcvB* (GcvB) or its derivatives. ΔR1: deletion mutant of the R1 seed region. ΔR1ΔR3: a combination of deletions of the R1 and R3 seed regions. mutR3, G156C, G160C, and C162G: nucleotide substitution mutations in the R3 seed region. GFP fluorescence of overnight-grown cells in LB medium was measured and divided by $OD_{600}$ for normalization. Base-pairing interaction between *aroG* mRNA (upper) and GcvB (below) was predicted by the IntaRNA program. Numbers above and below the nucleotide sequences indicate the nt location relative to the start codon of the mRNA and the transcription start site of GcvB, respectively. The start codon of *aroG* is shown in a box. (C) Relative mRNA levels quantified by qRT-PCR. *E. coli* Δ*gcvB*Δ*sroC* strain harboring pKP8-35 (vector) or pBAD-*gcvB* (GcvB) was grown in LB medium or M9 minimal medium supplemented with 0.2% glycerol. At exponential phase ($OD_{660} = 0.5$), 0.2% L-arabinose was added to the cultures to induce GcvB expression, and after 10 min, total RNA was isolated. Values are presented as mean ± standard error from three independent experiments ($n = 3$) and were statistically analyzed using one-way ANOVA with Bonferroni post hoc test in (A, B), or using the two-tailed Student's *t*-test in (C) (*$P < 0.05$, **$P < 0.01$, ***$P < 0.001$).

interactions within the R1 seed region (Table 1). The translational fusion of *aroL* was activated by ~1.6-fold in an R1-independent manner for an unknown reason.

To further confirm whether the levels of target mRNAs are affected by GcvB, we performed qRT-PCR analysis. The *E. coli* Δ*gcvB*Δ*sroC* strain was transformed by pBAD-*gcvB* and was grown in LB medium or M9 minimal medium supplemented with 0.2% glycerol until the optical density (OD) reached 0.5, and then the expression of GcvB

was induced by adding 0.2% L-arabinose for 10 min. We confirmed that the known GcvB targets, *dppA* and *map*, were significantly downregulated by the pulse expression of GcvB (Fig. 2C). It is noteworthy that the *map* mRNA level was dropped to ~30% despite only slight repression by GcvB at the translational level (24). Among the common pathway genes, only *aroG*, *aroC*, and *aroA* were significantly repressed by GcvB (Fig. 2C). Since *aroA* is cotranscribed with the upstream *serC* gene encoding the phosphoserine/phosphohydroxythreonine aminotransferase for serine and pyridoxal 5′-phosphate biosynthesis (30–32), GcvB is likely to regulate *serC* as the direct target. The 5′UTR of *serC* was predicted to interact strongly with the R1 seed of GcvB, and the translational fusion of *serC* was significantly repressed by GcvB in an R1-dependent manner (Fig. 2B). The levels of *serC* mRNA were affected by GcvB similarly to *aroA* (Fig. 2C). These results indicate that GcvB induces the degradation of *serC-aroA* bicistronic mRNA by directly targeting *serC*.

## Regulation of terminal biosynthetic pathways

The enzymes of Trp biosynthetic pathway are encoded by the *trp* operon composed of the *trpEDCBA* structural genes preceded by the *trpL* attenuator (33). The *trp* operon is transcribed mainly from the *trpL* promoter under the control of TrpR repressor, and the downstream genes are additionally transcribed from the internal *trpC* promoter (34, 35). We have previously found that GcvB represses the first structural gene *trpE* via the R1 seed region (24). To examine whether GcvB directly regulates the expression of

**TABLE 1** Summary of GcvB-mediated regulation of AAA biosynthetic and importer genes

| Gene | Reporter assay[a] | qRT-PCR % mRNA (*rne*[+] GcvB/ vector)[b] | | qRT-PCR % mRNA (*rne529* GcvB/vector)[b] | | IntaRNA[c] | |
|------|------------------|------------------|------|------------------|------|------------------|------|
| | % RFU/OD (GcvB/vector) | LB | M9 | LB | M9 | Hybridization energy (kcal/mol) | GcvB |
| *aroF* | 104 | 167 | 145 | | | −28.27 | 35–75 |
| *aroG* | 19.1 | 14.3 | 34.8 | 81.6 | 68.2 | −14.53 | 155–169 |
| *aroH* | 144 | 136 | 118 | | | −6.44 | 76–82 |
| *aroB* | 63.9 | 119 | 86.3 | | | −10.57 | 183–193 |
| *aroD* | 103 | 72 | 57.1 | | | −19.39 | 155–169 |
| *aroE* | 50.9 | 123 | 94.7 | | | −10.87 | 148–158 |
| *aroK* | 91.8 | 106 | 77.4 | | | −11.44 | 69–79 |
| *aroL* | 161 | 121 | 125 | | | −18.78 | 72–85 |
| *serC* | 62.3 | 40.1 | 48.2 | 69.4 | 71.5 | −17.42 | 72–84 |
| *aroA* | 124 | 45.7 | 39.9 | 78.3 | 75 | −14.31 | 63–73 |
| *aroC* | 47.8 | 28.7 | 29.5 | 59 | 23.5 | −21.97 | 76–90 |
| *pheA* | 35.4 | 27.8 | 89 | 85.9 | 87.1 | −18.97 | 63–93 |
| *tyrA* | 56 | 135 | 120 | | | −9.12 | 154–164 |
| *tyrB* | 55.6 | 135 | 95.8 | | | −24.61 | 31–64 |
| *trpE* | 20.3 | 46.5 | 71.5 | 86.9 | 120 | −20.28 | 72–91 |
| *trpD* | 84 | 54.5 | 48.5 | | | −17.88 | 54–69 |
| *trpC* | 49.9 | 78.8 | 56.5 | 109 | 102 | −16.81 | 77–90 |
| *trpB* | 82 | 84.7 | 81.9 | | | −17.22 | 63–82 |
| *trpA* | 65.8 | 100 | 77 | | | −5.6 | 72–78 |
| *aroP* | 13.6 | 15.1 | 15.7 | 93.6 | 25.1 | −17.36 | 71–91 |
| *pheP* | 58.1 | 79 | 59.1 | 119 | 71.4 | −11.34 | 66–78 |
| *tyrP* | 69 | 72.9 | 73.7 | | | −11.74 | 157–163 |
| *mtr* | 68.3 | 150 | 140 | | | −25.66 | 42–84 |
| *tnaC* | 107 | 51.7 | 91.6 | | | −6.45 | 163–169 |
| *tnaA* | 67.8 | 59.7 | 104 | | | −19.95 | 59–90 |
| *tnaB* | 27.5 | 72.2 | 87 | 158 | 161 | −20.8 | 52–72 |

[a]RFU/OD of the GcvB-expressing strain relative to the vector control measured by translational GFP reporter assay in this study and Reference (24).
[b]mRNA levels in the GcvB-expressing *rne* wild type and *rne529* strains relative to the vector control under LB and M9 minimal media measured by qRT-PCR in this study.
[c]Base-pairing interaction with GcvB predicted by IntaRNA. The GcvB-interacting region is presented in the right column "GcvB". Notably, nucleotide positions corresponding to the GcvB seed regions are as follows: R1: 65–93, R2: 129–145, R3: 150–172.

downstream genes, we constructed the translational fusions of intraoperonic regions using the pXG-30sf vector (26). Ectopic expression of GcvB significantly downregulated *trpC* and *trpA* in an R1-dependent manner (Fig. 3A). The IntaRNA program predicted the base pairing between the R1 seed region of GcvB and the coding region of *trpC*. However, according to the IntaRNA program, the interaction between the GcvB R1 seed and the *trpB-A* intergenic region cloned in the translational reporter plasmid was only seven base pairs including two wobble base pairs (Fig. 3A). qRT-PCR analysis confirmed that the mRNA levels of *trp* genes were elevated in the absence of Trp, and pulse expression of GcvB significantly downregulated the upstream three genes of *trp* operon but not *trpB* and *trpA* (Fig. 3C). This result is in line with the previous reports that the promoter-proximal segments of *trpEDCBA* mRNA is degraded faster than the promoter-distal segments (36, 37). We suggest that GcvB post-transcriptionally regulates the *trp* operon mainly by binding two different sites in *trpE* and *trpC* via the R1 seed region.

Previously, the MS2-affinity purification coupled with RNA sequencing (MAPS) analysis has revealed that GcvB significantly interacts with *pheA* and *tyrA* (20), both of which encode the first enzymes for Phe and Tyr biosynthetic pathways, respectively (5). A recent RNA interaction by ligation and sequencing (RIL-seq) study in an enteropathogenic *E. coli* has also detected the interaction between *pheA* and GcvB (38). The *tyrA* gene is located downstream of *aroF*, and the *aroF* promoter is transcriptionally regulated by TyrR in response to Tyr as a corepressor (39, 40). The transamination reaction to synthesize Phe and Tyr is preferentially catalyzed by TyrB, whose transcription is also under the control of TyrR (41). To verify whether GcvB regulates the Phe and Tyr biosynthetic pathways, the translational fusions of *pheA*, *tyrA*, and *tyrB* were constructed and examined by the reporter analysis. *pheA* was significantly repressed by GcvB, and as predicted, the R1 seed region is involved in the repression (Fig. 3B). GcvB moderately repressed *tyrA* and *tyrB* in an R1-dependent manner, but we could not find any base-pairing interactions within the R1 seed region (Table 1). qRT-PCR analysis showed that the pulse expression of GcvB decreased the mRNA levels of *pheA* exclusively in LB medium but had no effect in M9 medium, indicating a condition-specific regulation. In contrast, the mRNA levels of *aroF-tyrA* and *tyrB* were not significantly affected upon GcvB expression (Fig. 2C and 3C). Altogether, we suggest that GcvB represses the expression of first enzymes of the Trp and Phe terminal pathways but not that of Tyr (see the section "GcvB inhibits biosynthesis and import of AAAs").

## Regulation of AAA transporters

Among the five AAA transporters in *E. coli*, only *aroP* has previously been identified as the target of GcvB (24). Nonetheless, because GcvB redundantly regulates more than half of known amino acid transporters in *E. coli* (24), GcvB potentially interacts with mRNAs of the other AAA transporter genes. The R1 seed region is predicted to interact with *tnaB*, *pheP*, and *mtr*, but *tyrP* may only form a 7 bp hybrid with the R3 seed region (Table 1). The reporter analysis showed that GcvB strikingly repressed *tnaB* (28%) but only modestly *pheP*, *tyrP*, and *mtr* (Fig. 4A). qRT-PCR analysis verified significant downregulation of *pheP* and *tnaB* upon GcvB pulse expression during growth in M9 and LB media, respectively (Fig. 4C).

The polycistronic *tnaCAB* mRNA encodes the leader peptide, tryptophanase, responsible for the degradation of Trp into indole, and the Trp:H$^+$ symporter, respectively (42). It has been shown by the cross-linking, ligation, and sequencing of hybrids (CLASH) analysis that the *tnaCAB* mRNA interacts with GcvB at multiple sites (21). GcvB moderately repressed *tnaA* (68%), and deletion of R1 abrogated the repression (Fig. 4B). This result is in accordance with the R1-mediated interaction as predicted by IntaRNA. qRT-PCR analysis revealed that *tnaCAB* was highly expressed during growth in LB medium, and the mRNA levels at three coding regions of *tnaC*, *tnaA*, and *tnaB* were reduced by approximately twofold upon GcvB pulse expression (Fig. 4C). Northern blotting with a specific probe targeting *tnaA* mRNA identified two forms of the transcripts, *tnaCAB* (~3.1 kb) and *tnaCA* (~1.8 kb), and both transcripts were reduced

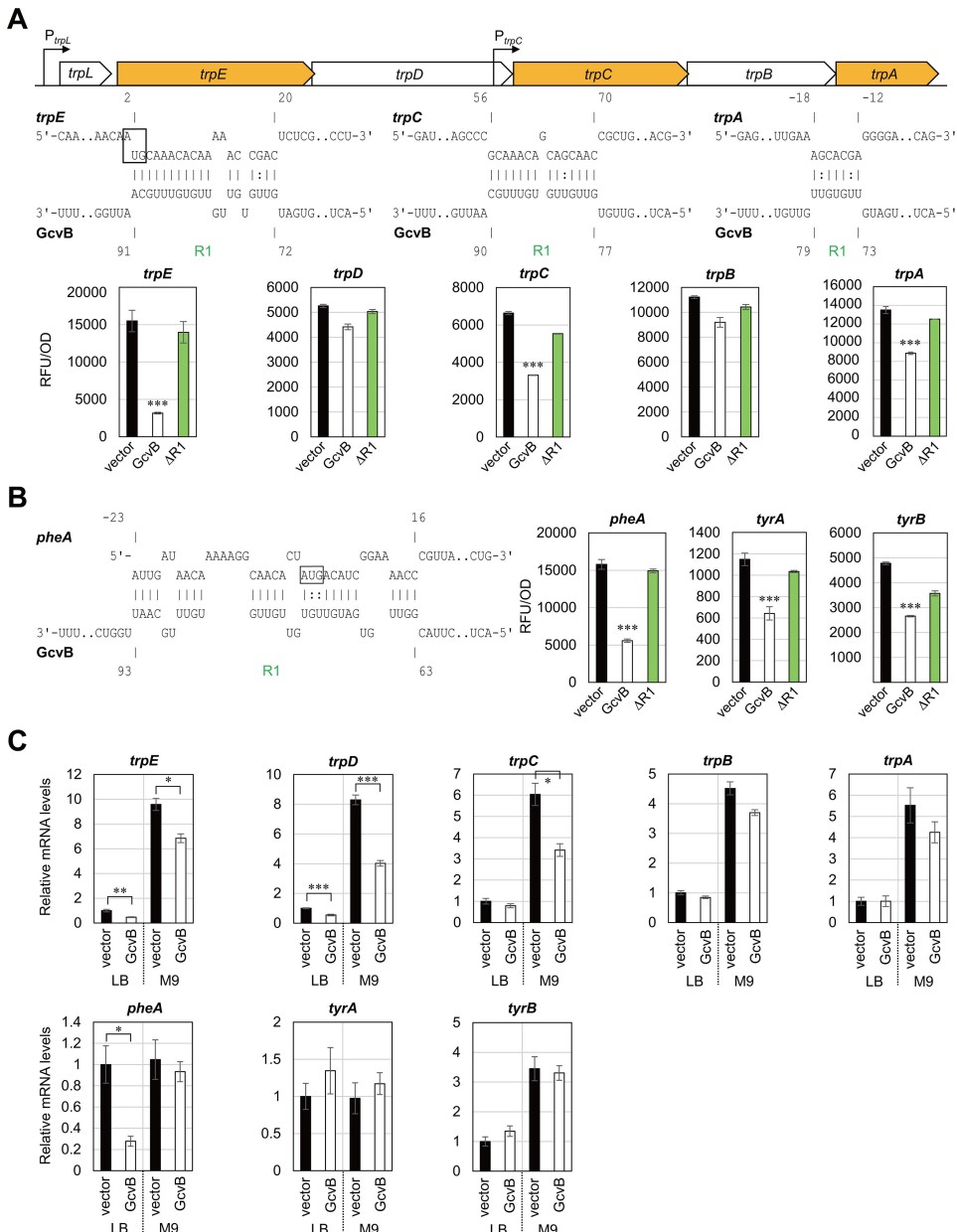

**FIG 3** Post-transcriptional regulation of the terminal biosynthetic pathways mediated by GcvB. (A, B) GFP reporter assays in *E. coli* Δ*gcvB*Δ*sroC* strain harboring pTP11 (vector), pP_L-*gcvB* (GcvB), or pP_L-*gcvB* ΔR1. GFP fluorescence of overnight-grown cells in LB medium was measured and divided by OD_600 for normalization. A schematic of the *trp* operon is provided above in (A). Base-pairing interactions between target mRNAs (upper) and GcvB (below) were predicted by the IntaRNA program. Numbers above and below the nucleotide sequences indicate the nt location relative to the start codon of the mRNA and the transcription start site of GcvB, respectively. The start codons of *trpE* and *pheA* are shown in a box. (C) Relative mRNA levels quantified by qRT-PCR as in Fig. 2C. Values are presented as mean ± standard error from three independent experiments (*n* = 3) and were statistically analyzed using one-way ANOVA with Bonferroni post hoc test in (A, B), or using the two-tailed Student's *t*-test in (C) (*$P < 0.05$, **$P < 0.01$, ***$P < 0.001$).

by approximately twofold upon GcvB pulse expression (Fig. 4D). These results suggest that GcvB induces the degradation of overall *tna* operon mRNA levels.

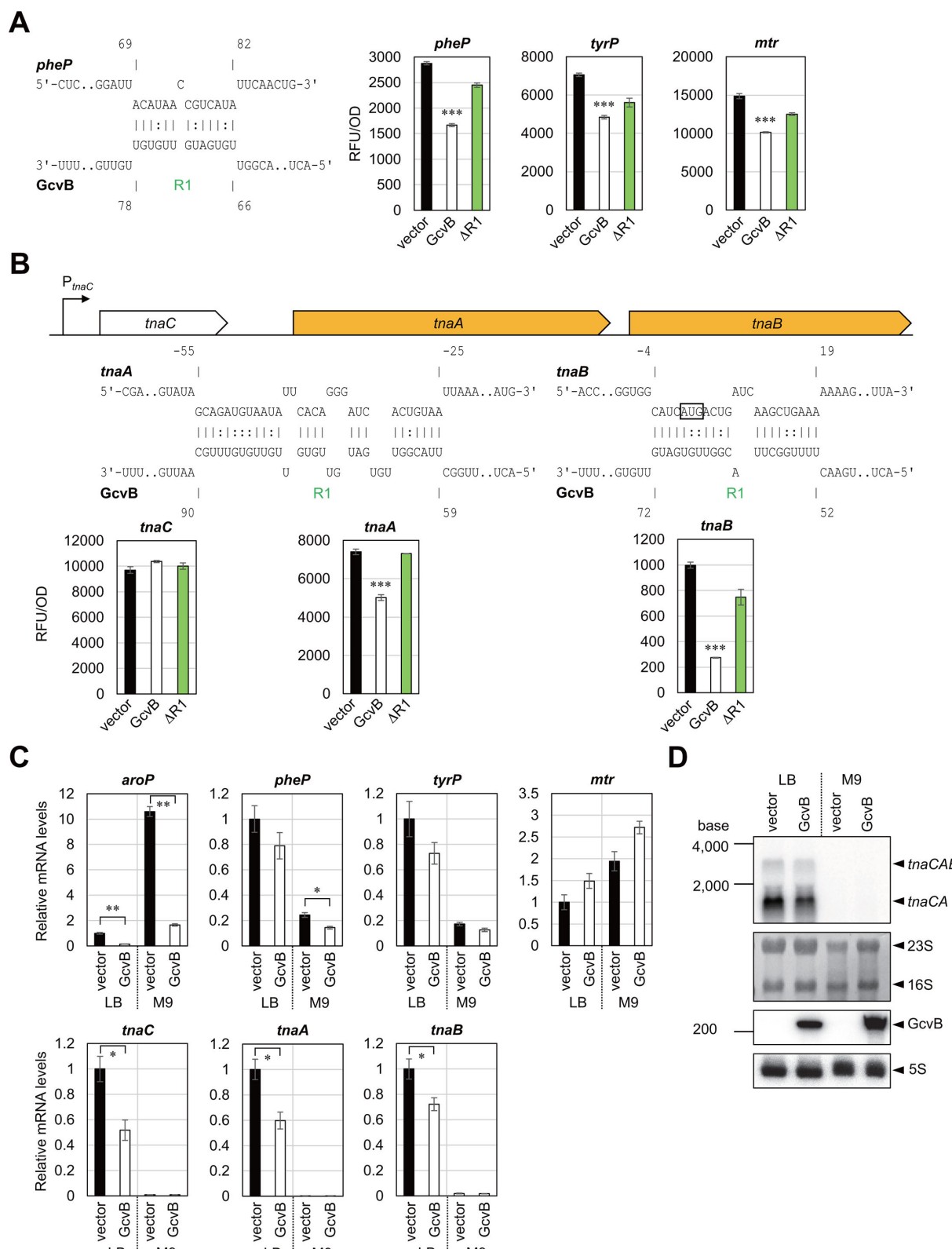

**FIG 4** Post-transcriptional regulation of the AAA transporters mediated by GcvB. (A, B) GFP reporter assays in *E. coli* Δ*gcvB*Δ*sroC* strain harboring pTP11 (vector), pP_L-*gcvB* (GcvB) or pP_L-*gcvB* ΔR1. GFP fluorescence of overnight-grown cells in LB medium was measured and divided by $OD_{600}$ for normalization. A schematic of the *tna* operon is provided above in (B). Base-pairing interactions between target mRNAs (upper) and GcvB (below) were predicted by the IntaRNA program. Numbers above and below the nucleotide sequences indicate the nt location relative to the start codon of the mRNA and the transcription start site of GcvB, (Continued on next page)

**Fig 4 (Continued)**

respectively. The start codon of *tnaB* is shown in a box. (C) Relative mRNA levels quantified by qRT-PCR as in Fig. 2C. Values are presented as mean ± standard error from three independent experiments ($n = 3$) and were statistically analyzed using one-way ANOVA with Bonferroni post hoc test in (A, B), or using the two-tailed Student's $t$-test in (C) (*$P < 0.05$, **$P < 0.01$, ***$P < 0.001$). (D) GcvB induces degradation of the *tnaCAB* mRNA. The same RNA samples as in (C) were analyzed by northern blotting. In each lane, 5 µg of total RNA was loaded and hybridized with [32]P-labeled antisense RNA probe targeting *tnaA* mRNA, GcvB sRNA, and 5S rRNA. 23S and 16S rRNA stained with methylene blue on the blot are indicated as loading controls.

## RNase E is responsible for the degradation of GcvB target mRNAs

It has been empirically shown that Hfq-dependent sRNAs exert post-transcriptional regulation primarily at the level of translation initiation, and in many cases, translational regulation accompanies secondary effects on the mRNA stability. Previous studies have shown that GcvB actively induces RNase E-mediated degradation of many target mRNAs (18, 20, 43).

The above results revealed a total of nine direct GcvB targets involved in AAA biosynthesis and transport, *aroG*, *aroC*, *serC*, *trpE*, *trpC*, *pheA*, *aroP*, *pheP*, and *tnaB*, while GcvB is likely to repress *aroA* indirectly. To verify whether RNase E is involved in the degradation of the target mRNAs upon GcvB pulse expression, we performed qRT-PCR analysis in the *rne598* background devoid of the C-terminal domain of RNase E (44), which interacts with Hfq and serves as a scaffold for sRNA-mediated target mRNA decay (45). Since GcvB itself is degraded by RNase E through interaction with SroC (27), we analyzed the target mRNA levels in the Δ*sroC* genetic background. As expected, the verified target mRNAs except *aroC* and *aroP* became insensitive to GcvB pulse expression (Fig. 5). This result indicates that RNase E is required for the decay of many GcvB targets but also implicates the other RNases in the post-transcriptional regulation of *aroC* and *aroP*.

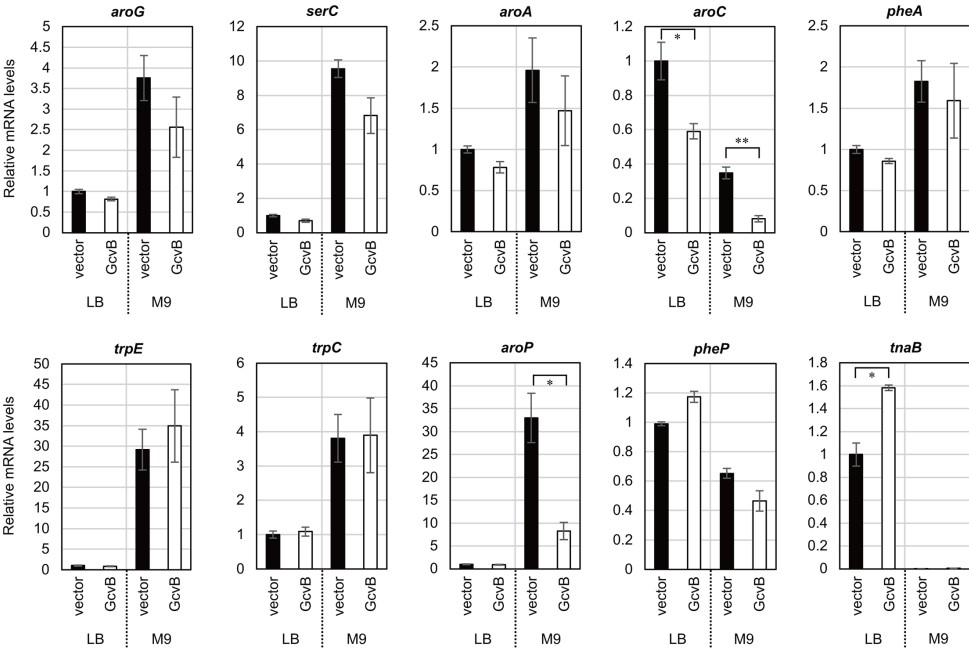

**FIG 5** Degradation of most target mRNAs of GcvB is dependent on RNase E. Relative mRNA levels quantified by qRT-PCR as in Fig. 2C, except that the *rne598* mutation was introduced as the background. Values are presented as mean ± standard error from three independent experiments ($n = 3$) and were statistically analyzed using the two-tailed Student's $t$-test (*$P < 0.05$, **$P < 0.01$).

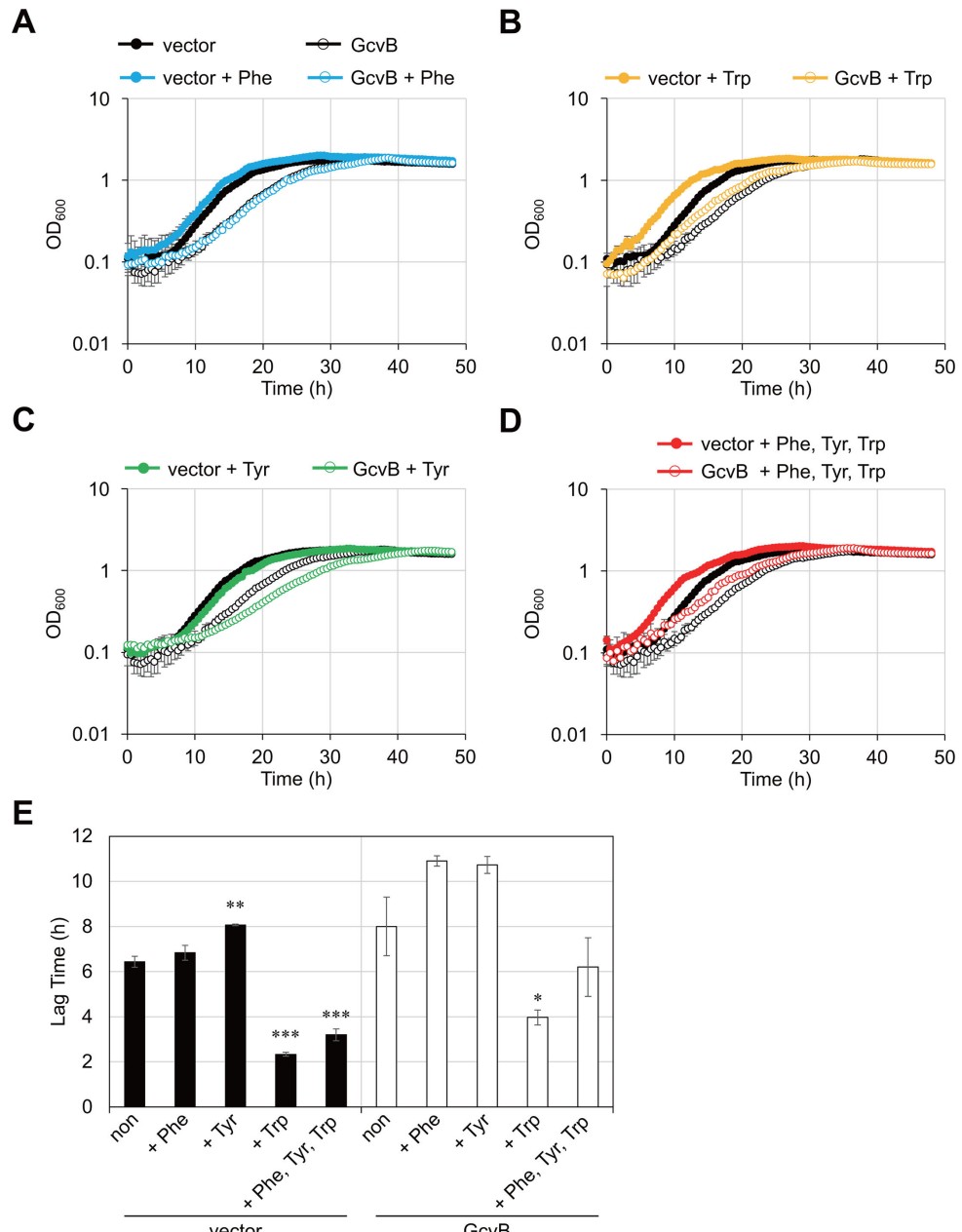

**FIG 6** GcvB induces growth defect by repressing import of AAAs under amino acid-limiting conditions. *E. coli ΔgcvBΔsroC* strain harboring pKP8-35 (vector) or pBAD-*gcvB* (GcvB) was grown in M9 minimal medium supplemented with 0.2% glycerol and 0.2% ʟ-arabinose in the presence of (A) Phe, (B) Trp, (C) Tyr, and (D) a combination of the three AAAs (1 mM). Growth was continuously monitored by measuring $OD_{600}$ every 30 min for 48 h. $OD_{600}$ values of the vector control and the GcvB-expressing strain in the absence of AAAs are shown as black closed circles and black open circles, respectively, as controls in (A–D). The calculated lag times based on these data are shown in (E). Data are presented as mean ± standard error from three independent experiments ($n = 3$) and were statistically analyzed using the one-way ANOVA with Bonferroni post hoc test (*$P < 0.05$, **$P < 0.01$, ***$P < 0.001$).

## GcvB inhibits biosynthesis and import of AAAs

Since GcvB represses the expression of several AAA biosynthetic enzymes and transporters, we hypothesized that overexpression of GcvB results in growth defects in a minimal medium. To test this, we compared the growth of *E. coli ΔgcvBΔsroC* strains harboring pBAD-*gcvB* or its vector control in M9 minimal medium supplemented with each AAA.

The addition of Phe or Trp improved the growth of the control strain (Fig. 6A and B). By contrast, Tyr posed a growth delay (Fig. 6C), which was completely restored by the addition of Phe and Trp (Fig. 6D). This result suggests that Tyr inhibits the other AAA synthetic enzymes.

The ectopic expression of GcvB generally extended the lag phase in the presence of AAAs (Fig. 6E), suggesting that GcvB inhibits the import of AAAs. Moreover, Tyr caused a more severe growth inhibition on the GcvB-expressing strain (Fig. 6C). The growth rate during exponential phase ($h^{-1}$) of the GcvB-expressing strain was 0.24 in the absence of Tyr and was decreased to 0.16 in its presence, although Tyr had no significant effect on the growth rate of the vector control (absence of Tyr: 0.33, presence of Tyr: 0.31). This result suggests that, in addition to the Tyr-driven feedback inhibition, GcvB inhibits the expression of AAA synthetic pathways other than the Tyr terminal pathway.

Azaserine (*O*-diazoacetyl-L-serine), an antibiotic compound produced by *Streptomyces fragilis*, is taken up through AroP and the ABC transporter LivJKHMGF to inhibit the growth of *E. coli* (46, 47). To test whether GcvB alleviates the toxicity of azaserine by limiting its import, we determined the minimal inhibitory concentration (MIC) of *E. coli* Δ*gcvB*Δ*sroC* strains harboring pP$_L$-*gcvB* or its vector control in M9 minimal medium containing 0.4% glucose. The MIC of azaserine was 30 µM in the strain harboring the control vector, and the GcvB-expressing strain exhibited a twofold higher MIC of 60 µM. Altogether, these results indicate that GcvB inhibits the uptake of AAAs.

## DISCUSSION

This study shows that GcvB post-transcriptionally regulates AAA metabolism at several key steps (Fig. 7). Together with our previous report (24), we conclude that GcvB represses a total of six biosynthetic genes (*aroG*, *aroC*, *serC-aroA*, *trpE*, *trpC*, and *pheA*) and three transporter genes (*aroP*, *pheP*, and *tnaB*) as summarized in Table 1. Many of these target genes are downregulated via the conserved R1 seed region of GcvB. Nonetheless, GcvB interacts with the translation initiation region of *aroG* mRNA with the R3 seed region to repress the expression of the first enzyme of the common pathway. For *aroB*, *aroE*, *aroL, trpA*, *tyrA*, *tyrB*, *tyrP*, and *mtr*, the expression levels of translational fusions were affected by GcvB overexpression, but their mRNA levels were not significantly downregulated (Table 1). It is possible that these genes are regulated by GcvB mainly at the translational level, but the predicted base-pairing regions are not located near the R1 seed region. GcvB might possess sequences other than the known three conserved seeds to interact with the target mRNAs.

We observe a substantial overlap between the transcriptional and post-transcriptional regulons of amino acid metabolism in *E. coli* (9). Both TyrR and GcvB repress *aroG* and *aroP*, which encode the major DAHP synthase and the major AAA transporter, respectively, while TrpR and GcvB together repress the *trp* operon (Fig. 7). Interestingly, the Trp and Phe biosynthetic genes are regulated at the post-transcriptional level both by the small RNA and the transcriptional attenuators, while the Tyr biosynthetic genes are predominantly regulated at the transcriptional level. In addition, the global transcriptional regulator Lrp regulates one-third of the *E. coli* genome, but *aroG* is the only direct target of Lrp among the AAA metabolic genes (48). Lrp activates the transcription of *aroG* in the absence of leucine and dissociates from the promoter in the presence of leucine (49). We demonstrate that the translation of *aroG* is directly repressed by GcvB via the R3 seed sequence (Fig. 2A). To control the biosynthesis of AAAs, the first reaction of the common pathway is critical. Since AroG is the major isozyme and is allosterically inhibited by Phe, a *gcvB* mutation combined with the feedback resistance mutation in *aroG* (50) will be beneficial for the fermentative production of AAAs. By contrast, the last reaction of the common pathway is also rate-limiting. However, the expression of *aroC* is not regulated by any transcriptional regulators and has long been believed to be constitutive in *E. coli* (5). We have previously reported that GcvB downregulates the expression of *aroC* in the *prmB-aroC-mepA-yfcA-epmC-yfcL* gene cluster, leaving the expression level of *prmB* constant (24). Therefore, we

conclude that GcvB is the crucial sRNA regulator of AAA biosynthesis by targeting the first and last steps in the common pathway.

Post-transcriptional regulation of AAA biosynthetic pathways extends through two additional sRNAs. An Hfq-dependent sRNA RydC primarily activates the expression of cyclopropane fatty acid synthase by stabilizing the *cfa* mRNA (51) and modestly regulates *trpE* and *pheA* using different seed sequences (52). The expression of RydC is regulated by a GntR family transcriptional repressor YieP, whose signaling molecule is unknown (53). An Hfq-independent sRNA RybA is induced by peroxide stress and represses *aroF* and *aroL* in a TyrR-dependent manner without affecting the level of *tyrR* mRNA (54). RybA encodes a small peptide MntS, which is involved in manganese homeostasis by inhibiting the MntP transporter (55, 56). Further study is required to understand how significantly these sRNAs along with GcvB regulate AAA metabolism and correlate it with the other biological pathways.

Identifying the direct targets of sRNAs is important to understand the impact of post-transcriptional regulation through the base-pairing mechanism. GcvB is widely conserved in *Enterobacteriaceae* and the other families of γ-Proteobacteria and has been regarded as one of the model sRNAs targeting multiple mRNAs. Pioneering multi-omics studies in *Salmonella* revealed that GcvB specifically represses the substrate-binding components of amino acid ABC transporters among the proteome of periplasmic

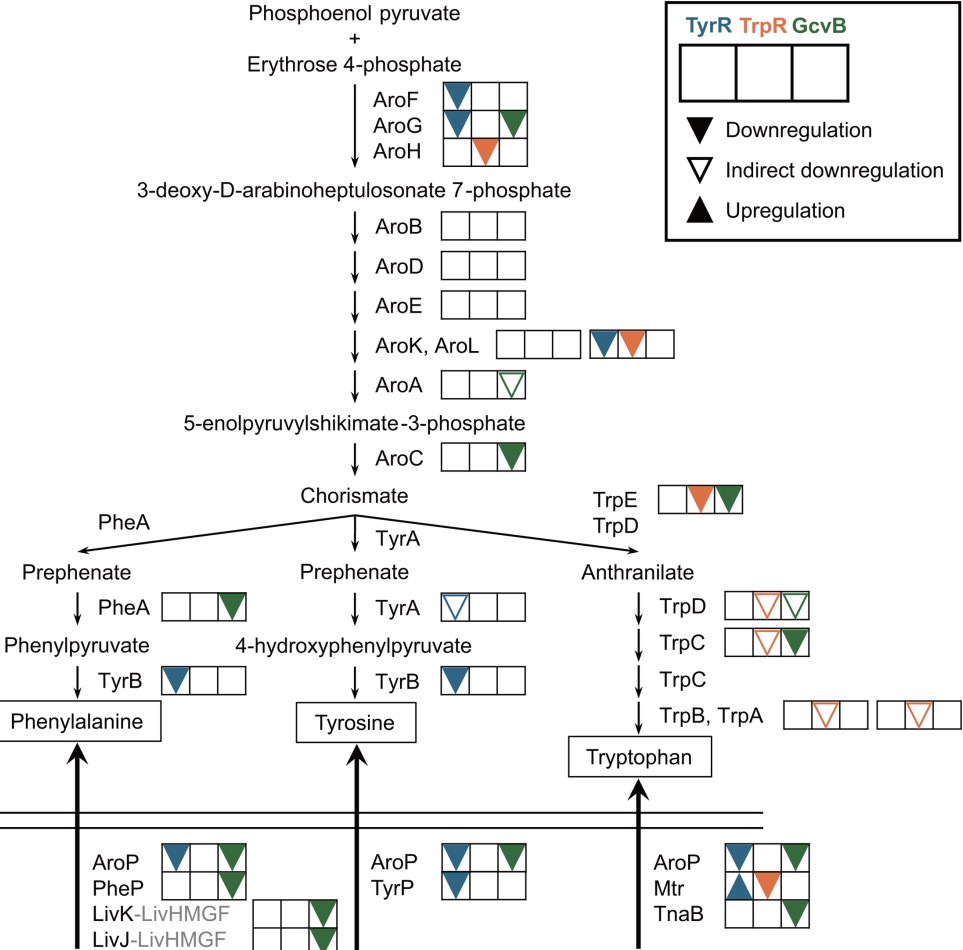

**FIG 7** Schematic representation of the transcriptional and post-transcriptional regulation of the AAA biosynthesis and import pathways in *E. coli*. Transcriptional regulation by TyrR and TrpR and post-transcriptional regulation by GcvB are shown next to the corresponding enzymes. ▼: direct downregulation, ▽: indirect downregulation via upstream gene regulation, ▲: direct upregulation.

fraction and alters the transcriptome as translation inhibition induces degradation of most target mRNAs (11, 12). In addition, a bioinformatic search allowed to discover many interactions with the G/U-rich R1 seed sequence of GcvB (12). Recently, base-pairing interactions between RNAs can be experimentally identified through advanced transcriptomic techniques (57, 58). We have previously compiled the RNA–RNA interactome data sets to identify new GcvB targets, raising the number of validated GcvB targets to more than 50 (24). However, as *aroG* was predicted as a target through a bioinformatic approach (59) but did not fit the criteria adopted in our previous survey (24), genuine GcvB targets are still underrepresented in the previous RNA–RNA interactome data sets due to low expression levels. This study reveals new sRNA targets by thoroughly examining genes of particular metabolic functions and thus highlights the importance of laborious reporter assays dedicated to sRNA-mediated post-transcriptional regulation.

The criteria to discriminate between targets and non-targets of sRNAs can be somewhat arbitrary. The previously identified GcvB target, *map*, is translationally repressed to ~70% via the R3 seed region (24), but its mRNA level is strikingly downregulated to 30% (Fig. 2C). By contrast, the mRNA levels of *aroB, aroE, trpA, tyrA,* and *tyrB* were not significantly altered upon GcvB pulse expression, but their translational fusions were substantially repressed (Table 1), suggesting that these genes might be regulated solely at the translational level. This type of sRNA targets cannot be identified merely by transcriptomics but by proteomics or ribosome profiling (Ribo-seq) (60), as represented by the translational regulation of *lpp* by MicL sRNA (61). Ribo-seq analysis has revealed that RyhB sRNA regulates >50 genes involved in iron utilization but does not always alter the level of target mRNAs (62). Interestingly, although GcvB abundantly interacts with the *raiA* mRNA encoding the ribosome-associated inhibitor (20–24), Faigenbaum-Romm et al. have shown that GcvB overexpression exerts no regulatory output on the *raiA* mRNA, arguing that a subset of the RNA–RNA interactions are sporadic (63). To complete the identification of sRNA regulon, we require more RNA–RNA interactome data sets and expression profiles at both transcription and translation levels under various growth conditions relevant to the sRNA of interest and further validation in physiological significance using standard methods.

## MATERIALS AND METHODS

### Bacterial strains and growth media

The strains used in this study are listed in Table 2. Bacterial cells were grown at 37°C with reciprocal shaking at 180 rpm in LB Miller medium (Nacalai Tesque) or M9 minimal medium supplemented with 0.2% glycerol. Where required, media were supplemented with antibiotics at the following concentrations: 50 µg/mL ampicillin, 50 µg/mL kanamycin, and 12.5 µg/mL chloramphenicol. To construct the Δ*gcvB*Δ*sroC rne598* strain, the *rne598*-FLAG-*cat* allele was transduced using P1 phage from strain TM529 (44) into the Δ*gcvB*Δ*sroC* strain. The *cat* cassette was removed by introducing pCP20, followed by curation during growth at 37°C.

**TABLE 2**  Bacterial strains used in this study

| Strain | Relevant markers/ genotype | Reference/ source |
|---|---|---|
| BW25113 | F⁻ λ⁻ *rrnB3* Δ*lacZ4787 hsdR514* Δ(*araBAD*)567 Δ(*rhaBAD*)568 *rph*-1 | NBRP strain |
| Δ*gcvB*Δ*sroC* | BW25113 Δ*gcvB*::*kan* Δ*sroC*::FRT | (24) |
| TM529 | W3110 *mlc rne598*-FLAG-*cat* | (44) |
| Δ*gcvB*Δ*sroC rne598* | Δ*gcvB*Δ*sroC rne598*-FLAG-FRT | This study |

## Oligonucleotides and plasmids

The plasmids and oligonucleotides used in this study are listed in Tables S1 and S2, respectively. The plasmids for constitutive expression of GcvB (pP$_L$-*gcvB*) or its derivatives were constructed previously (24). The L-arabinose-inducible expression plasmid for *E. coli* GcvB (pBAD-*gcvB*) was constructed using the primers JVO-0895 and MMO-0086 as previously described (12). Translational fusions were constructed as previously described (25, 26).

## GFP fluorescence quantification

The *E. coli* BW25113 Δ*gcvB*Δ*sroC* strains harboring a combination of the *sfgfp* translational fusions and pP$_L$-*gcvB* or its derivatives were inoculated from single colonies into 400 µL LB medium in 96 deep-well plates (Thermo Scientific) and were grown overnight at 37°C with rotary shaking at 1,200 rpm in DWMax M-BR-032P plate shaker (Taitec). A 100 µL aliquot of the overnight cultures was transferred into 96-well optical bottom black microtiter plates (Thermo Scientific), and both optical density at 600 nm (OD$_{600}$) and fluorescence (excitation at 485 nm and emission at 535 nm with dichroic mirror of 510 nm, fixed gain value of 50) were measured using Spark plate reader (Tecan). The relative fluorescence unit (RFU) was calculated by subtracting the autofluorescence of bacterial cells of the same strains harboring the GcvB-expressing plasmids.

## Total RNA extraction

The *E. coli* BW25113 Δ*gcvB*Δ*sroC* strains harboring pBAD-*gcvB* or its vector control (pKP8-35) (64) were grown in LB medium or M9 minimal medium supplemented with 0.2% glycerol. When the OD$_{660}$ reached 0.5, L-arabinose was added to the cultures at a final concentration of 0.2% to induce GcvB expression. After 10 min, two volumes of RNA Protect Bacterial Reagent (Qiagen) was added to one volume of the cultures to stabilize cellular RNA. Total RNA was isolated using NucleoSpin RNA (Macherey–Nagel) according to the manufacturer's instruction. The RNA samples were treated with DNase I at room temperature for 30 min.

## qRT-PCR

cDNA was synthesized using ReverTra Ace qPCR RT Master Mix with gDNA Remover (Toyobo). qRT-PCR was performed using TB Green Premix Ex Taq™ II (Takara Bio) on QuantStudio 5 Real-Time PCR System (Thermo Scientific). Each target-gene mRNA level was normalized to a reference gene transcript (16S rRNA) from the same RNA sample. Fold changes were determined using the $2^{-\Delta\Delta Ct}$ method (65). The sequences of the primers used are shown in Table S2.

## Northern blotting

For mRNA analysis, total RNA (5 µg) was separated by 1.5% agarose gel electrophoresis in the presence of formaldehyde. DynaMarker Prestain Marker for RNA high (BioDynamics Laboratory) was used as a size marker. The gels were transferred onto Nytran SuPerCharge nylon membranes (Cytiva) overnight by capillary blotting using TurboBlotter Kit (Cytiva). The membranes were crosslinked with transferred RNA by UV light at 120 mJ/cm$^2$. Before probe hybridization, the membranes were stained with 0.3 M sodium acetate containing 0.03% methylene blue. A 161 bp [$^{32}$P]-labeled antisense RNA probe targeting *tnaA* mRNA was synthesized by *in vitro* transcription using the MAXIscript kit (Invitrogen). The specific DNA fragments required for probe generation were amplified by PCR with the primers MMO-1062 and MMO-1704 (Table S2). Prehybridization (1 h) and hybridization (24 h) were performed at 70°C in ULTRAhyb ultrasensitive hybridization buffer (Thermo Scientific). The membranes were subsequently washed in 2 × SSC/0.1% SDS at 20°C for 20 min, 1 × SSC/0.1% SDS at 65°C for 15 min, and 0.5 × SSC/0.1% SDS at 65°C for 15 min.

For sRNA analysis, total RNA was isolated using the TRIzol reagent (Invitrogen) and precipitated with isopropanol and cold ethanol. RNA was quantified using NanoDrop One (Invitrogen). Total RNA (5 µg) was separated by 6% polyacrylamide/7 M urea gel electrophoresis in $1 \times$ TBE buffer. DynaMarker RNA Low II ssRNA fragments (BioDynamics Laboratory) was used as a size marker. The gels were transferred onto Hybond-XL nylon membrane (GE Healthcare) by electroblotting. The membrane was crosslinked with transferred RNA by 120 mJ/cm$^2$ UV light, incubated for prehybridization in Rapid-Hyb buffer (Amersham) at 42°C for 1 h, and then incubated for hybridization with a [$^{32}$P]-labeled probe JVO-0750 and MMO-1056 at 42°C overnight to detect GcvB and 5S rRNA, respectively. The membrane was washed in $5 \times$ SSC/0.1% SDS at 42°C for 15 min, $1 \times$ SSC/0.1% SDS at 42°C for 15 min, and $0.5 \times$ SSC/0.1% SDS at 42°C for 15 min.

The membranes were exposed to imaging plates (Fujifilm), and the resulting signals were visualized using Typhoon FLA7000 scanner (GE Healthcare) and quantified using Image Quant TL software (GE Healthcare).

## Growth assay

The *E. coli* BW25113 Δ*gcvB*Δ*sroC* strains harboring pBAD-*gcvB* or its vector control (pKP8-35) were grown overnight in LB medium. The cells were harvested by centrifugation (5,000×*g*, 20°C, 5 min), washed twice with M9 minimal medium, and diluted to 1:100 in M9 minimal medium supplemented with 0.2% glycerol and 0.2% L-arabinose in the presence or absence of Phe, Tyr, or Trp (final concentration: 1 mM). A 100 µL aliquot of the cell suspension was transferred into 96-well flat-bottom clear microtiter plates (Iwaki). The plates were incubated at 37°C with rotary shaking at 180 rpm with a 3 mm amplitude for 48 h in Humidity Cassette using Spark plate reader (Tecan). Growth was continuously monitored by measuring $OD_{600}$ every 30 min. Lag time was calculated using microbial lag phase duration calculator (66).

## Determination of MIC of azaserine

Biological triplicates of the *E. coli* BW25113 Δ*gcvB*Δ*sroC* strains harboring pP$_L$-*gcvB* or its vector control (pTP11) were grown overnight in LB medium. The cells were harvested by centrifugation (5,000×*g*, 20°C, 5 min), washed twice with M9 minimal medium, and diluted 1:100 in M9 minimal medium supplemented with 0.4% glucose in the presence or absence of azaserine (final concentration: 20–100 µM). A 100 µL aliquot of the cell suspension was transferred into 96-well flat-bottom clear microtiter plates (Iwaki). The plates were incubated at 37°C with rotary shaking at 180 rpm with a 3 mm amplitude for 72 h in Humidity Cassette using Spark plate reader (Tecan). Growth was continuously monitored by measuring $OD_{600}$ every 1 h.

## Statistical analysis

The significance of the results was calculated using the two-tailed Student's *t*-test or one-way ANOVA with the Bonferroni post hoc test (*$P < 0.05$, **$P < 0.01$, ***$P < 0.001$).

## ACKNOWLEDGMENTS

The authors would like to thank Teppei Morita at Keio University and NBRP-*E.coli* at NIG for providing *E. coli* strains.

This study was supported by JSPS KAKENHI grant numbers JP22K14809 and JP22KJ0376 to T.K. and JP24K01661 to M.M. T.K. is supported by JSPS Postdoctoral Fellowship, IFO scholarship for young researchers (Y-2022-2-028), and Kato Memorial Bioscience Foundation (2023B-101). Work in the Miyakoshi laboratory is supported by Mishima Kaiun Memorial Foundation, Asahi Group Foundation, and Takeda Science Foundation.

## AUTHOR AFFILIATIONS

[1]Department of Infection Biology, Institute of Medicine, University of Tsukuba, Ibaraki, Japan

[2]Transborder Medical Research Center, University of Tsukuba, Ibaraki, Japan

[3]International Joint Degree Master's Program in Agro-Biomedical Science in Food and Health (GIP-TRIAD), University of Tsukuba, Ibaraki, Japan

## AUTHOR ORCIDs

Takeshi Kanda http://orcid.org/0000-0003-3587-5037
Masatoshi Miyakoshi http://orcid.org/0000-0002-4901-2809

## FUNDING

| Funder | Grant(s) | Author(s) |
| --- | --- | --- |
| MEXT \| Japan Society for the Promotion of Science (JSPS) | JP22K14809 | Takeshi Kanda |
| MEXT \| Japan Society for the Promotion of Science (JSPS) | JP22KJ0376 | Takeshi Kanda |
| MEXT \| Japan Society for the Promotion of Science (JSPS) | JP24K01661 | Masatoshi Miyakoshi |
| Institute for Fermentation, Osaka (IFO) | Y-2022-2-028 | Takeshi Kanda |
| Kato Memorial Bioscience Foundation | 2023B-101 | Takeshi Kanda |
| Mishima Kaiun Memorial Foundation | | Masatoshi Miyakoshi |
| Asahi Group \| Asahi Group Foundation (Asahi Group Research Foundation) | | Masatoshi Miyakoshi |
| Takeda Science Foundation (TSF) | | Masatoshi Miyakoshi |

## AUTHOR CONTRIBUTIONS

Takeshi Kanda, Data curation, Formal analysis, Funding acquisition, Investigation, Methodology, Validation, Visualization, Writing – original draft, Writing – review and editing | Toshiko Sekijima, Formal analysis, Investigation, Validation, Writing – review and editing | Masatoshi Miyakoshi, Conceptualization, Funding acquisition, Investigation, Methodology, Project administration, Resources, Supervision, Validation, Writing – original draft, Writing – review and editing

## ADDITIONAL FILES

The following material is available online.

### Supplemental Material

**Supplemental material (Spectrum02035-24-S0001.pdf).** Tables S1 to S4.

### Open Peer Review

**PEER REVIEW HISTORY (review-history.pdf).** An accounting of the reviewer comments and feedback.

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
