## [Reviewer comments · Microbiology Spectrum]

Microbiology Spectrum

Post-transcriptional regulation of aromatic amino acid metabolism by GcvB small RNA in *Escherichia coli*

Takeshi Kanda, Toshiko Sekijima, and Masatoshi Miyakoshi

Corresponding Author(s): Masatoshi Miyakoshi, Tsukuba Daigaku

Review Timeline:

Submission Date:	September 30, 2024
Editorial Decision:	November 19, 2024
Revision Received:	December 2, 2024
Editorial Decision:	December 11, 2024
Revision Received:	December 12, 2024
Accepted:	December 16, 2024

Editor: Tino Polen

Reviewer(s): Disclosure of reviewer identity is with reference to reviewer comments included in decision letter(s). The following individuals involved in review of your submission have agreed to reveal their identity: Tanmay Dutta (Reviewer #1)

Transaction Report:

DOI: <https://doi.org/10.1128/spectrum.02035-24>

Re: Spectrum02035-24 (Post-transcriptional regulation of aromatic amino acid metabolism by GcvB small RNA in *Escherichia coli*)

Dear Dr. Masatoshi Miyakoshi,

thank you for submitting your manuscript to Microbiology Spectrum.

The expert reviewers suggest modifications which I do hope you can take into account in a revised version.

Please return the revised manuscript within 60 days; if you cannot complete the modification within this time period, please contact me. If you do not wish to modify the manuscript and prefer to submit it to another journal, notify me immediately so that the manuscript may be formally withdrawn from consideration by Spectrum.

Revision Guidelines

Sincerely,
Tino Polen
Editor
Microbiology Spectrum

Reviewer #1 (Comments for the Author):

The manuscript entitled "Post-transcriptional regulation of aromatic amino acid metabolism by GcvB small RNA in *Escherichia coli*" by Kanda et al. delineates a small RNA GcvB-mediated regulation of the expression of several genes associated with amino acid biosynthesis and transport. A combination of translation fusion and qRT-PCR was performed to examine the expression. This is a continuation of a previous study by the authors. Overall, the authors concluded that several new GcvB targets, some of which are either AAA biosynthetic genes or AAA transporters. All in all, the work is not sufficient to clearly draw the mechanistic conclusions although the manuscript is in much better shape after revision.

A few major points must be clarified.

1. Although in the response to the reviewer section author mentioned that the R2 seed region was never tested as the candidate genes in this study were not predicted to interact with R2. However, the authors showed that ectopic expression of GcvB downregulates *trpA* in an R1-dependent manner although the IntaRNA program did not predict any significant base pairing between them (lines 144-149). IntaRNA prediction also has limitations and thus, the contribution of the R2 seed region should have been tested.
2. Authors should have created a *gcvB* deletion mutant and a complementary strain overexpressing GcvB. Comparing the expression studies in the wild type, $\Delta gcvB$, and complementary strain would have given clear mechanistic detail.
3. It is hard to interpret anything from the growth curve (Fig 6) as these are presented in linear scale. How the lag time was calculated.
4. The quality of Fig 7 is really bad.

Reviewer #2 (Comments for the Author):

The manuscript by Kanda et al. focused on identifying and validating targets of the sRNA GcvB that are involved in aromatic amino acid (AAA) biosynthesis and uptake. The authors built on previous work that reported *aroC*, *aroP*, and *trpE* as GcvB targets and utilized GFP translational fusions and qRT-PCR to identify new GcvB targets involved in these processes. IntaRNA was used to predict pairing regions between GcvB and its targets, however (apart from *aroG*) the investigation of the role of seed regions in regulation was limited to the effects of seed region R1. Although additional experiments into the mechanisms of GcvB regulation would have elevated the manuscript, it expands upon the current understanding of AAA regulation and may be beneficial in the utilization of *Escherichia coli* for the fermentative production of AAAs.

1. Explain the construction of your translational fusions.
2. Lines 117-119. How do you know that another seed region is not involved in the pairing and regulation like you saw with *aroG*?
3. Line 171. Why do you say that GcvB is not repressing the expression of the first enzymes of the Tyr pathway when your GFP reporter showed moderate repression of both *tyrA* and *tyrB*?
4. Line 210. Inconsistent how?
5. Line 217. How can you say "significant" without statistics?
6. Table 2. What does the red indicate?
7. Figure 4D. Please include GcvB expression levels between vector and +GcvB in your northern?

The manuscript entitled “Post-transcriptional regulation of aromatic amino acid metabolism by GcvB small RNA in Escherichia coli” by Kanda et al. delineates a small RNA GcvB-mediated regulation of the expression of several genes associated with amino acid biosynthesis and transport. A combination of translation fusion and qRT-PCR was performed to examine the expression. This is a continuation of a previous study by the authors. Overall, the authors concluded that several new GcvB targets, some of which are either AAA biosynthetic genes or AAA transporters. All in all, the work is not sufficient to clearly draw the mechanistic conclusions although the manuscript is in much better shape after revision.

A few major points must be clarified.

1. Although in the response to the reviewer section author mentioned that the R2 seed region was never tested as the candidate genes in this study were not predicted to interact with R2. However, the authors showed that ectopic expression of GcvB downregulates *trpA* in an R1-dependent manner although the IntaRNA program did not predict any significant base pairing between them (lines 144-149). IntaRNA prediction also has limitations and thus, the contribution of the R2 seed region should have been tested.
2. Authors should have created a *gcvB* deletion mutant and a complementary strain overexpressing GcvB. Comparing the expression studies in the wild type, $\Delta gcvB$, and complementary strain would have given clear mechanistic detail.
3. It is hard to interpret anything from the growth curve (Fig 6) as these are presented in linear scale. How the lag time was calculated.
4. The quality of Fig 7 is really bad.

Reviewer #1 (Comments for the Author):

The manuscript entitled "Post-transcriptional regulation of aromatic amino acid metabolism by GcvB small RNA in Escherichia coli" by Kanda et al. delineates a small RNA GcvB-mediated regulation of the expression of several genes associated with amino acid biosynthesis and transport. A combination of translation fusion and qRT-PCR was performed to examine the expression. This is a continuation of a previous study by the authors. Overall, the authors concluded that several new GcvB targets, some of which are either AAA biosynthetic genes or AAA transporters. All in all, the work is not sufficient to clearly draw the mechanistic conclusions although the manuscript is in much better shape after revision. A few major points must be clarified.

1. Although in the response to the reviewer section author mentioned that the R2 seed region was never tested as the candidate genes in this study were not predicted to interact with R2. However, the authors showed that ectopic expression of GcvB downregulates trpA in an R1-dependent manner although the IntaRNA program did not predict any significant base pairing between them (lines 144-149). IntaRNA prediction also has limitations and thus, the contribution of the R2 seed region should have been tested. We agree that IntaRNA prediction has limitations. In the previous manuscript, the coding region of trpA mRNA was predicted to interact with a GcvB sequence proximal to the R1 region. To clarify this base-pairing is critical for the post-transcriptional regulation by GcvB, we shortened the trpA coding region from 90 to 30 nucleotides to construct the new translational reporter. However, the short translational fusion was still repressed by GcvB at a moderate level, ruling out the possibility that the predicted region was involved in the repression. Now we find a weak interaction between trpA translation initiation region and GcvB R1. We have rewritten Lines 147-149 and modified Fig 3A and Table 1 in the revised manuscript.
2. Authors should have created a gcvB deletion mutant and a complementary strain overexpressing GcvB. Comparing the expression studies in the wild type, Δ gcvB, and complementary strain would have given clear mechanistic detail. Thank you for your suggestion. As replied to the previous comments, the global expression changes in a gcvB deletion strain and its complementary strain may include many indirect effects because GcvB regulates the expression of global transcriptional regulators such as Lrp, CsgD, and PhoP. Nonetheless, we will analyze the global expression changes with these strains grown in relevant conditions in our follow-up study.
3. It is hard to interpret anything from the growth curve (Fig 6) as these are presented in linear scale. How the lag time was calculated.

The previous version transferred to Microbiology Spectrum had already been modified into log scale. As written in the Methods section, lag time was calculated using Microbial lag phase duration calculator (Smug et al. 2024. *Methods Ecol Evol* 15:301–307).

4. The quality of Fig 7 is really bad.

Fig 7 is omitted in the revised manuscript.

Reviewer #2 (Comments for the Author):

The manuscript by Kanda et al. focused on identifying and validating targets of the sRNA GcvB that are involved in aromatic amino acid (AAA) biosynthesis and uptake. The authors built on previous work that reported *aroC*, *aroP*, and *trpE* as GcvB targets and utilized GFP translational fusions and qRT-PCR to identify new GcvB targets involved in these processes. IntaRNA was used to predict pairing regions between GcvB and its targets, however (apart from *aroG*) the investigation of the role of seed regions in regulation was limited to the effects of seed region R1. Although additional experiments into the mechanisms of GcvB regulation would have elevated the manuscript, it expands upon the current understanding of AAA regulation and may be beneficial in the utilization of *Escherichia coli* for the fermentative production of AAAs.

1. Explain the construction of your translational fusions.

We have added the explanation in Lines 94-96.

2. Lines 117-119. How do you know that another seed region is not involved in the pairing and regulation like you saw with *aroG*?

We have predicted the interaction between *aroE* and the R3 region of GcvB (Table 1). However, deletion of R3 did not affect the translational repression of *aroE* (data not shown). The predicted interaction region in *aroB* is located in the terminator of GcvB and was not tested in detail. In order to avoid the confusion, this sentence was omitted in the revised manuscript.

3. Line 171. Why do you say that GcvB is not repressing the expression of the first enzymes of the Tyr pathway when your GFP reporter showed moderate repression of both *tyrA* and *tyrB*?

As written in the section "GcvB inhibits biosynthesis and import of AAAs", the addition of Tyr into the M9 minimal medium caused a growth inhibition due to negative feedback inhibition of Trp and Phe biosynthesis, and overexpression of GcvB further inhibited the growth (Fig. 6C). In contrast, the ectopic expression of GcvB did not significantly affect the growth in M9 medium in the presence of Phe (Fig. 6A). This suggests that GcvB does not repress the biosynthesis of Tyr. Accordingly, we have added a note in the end of this

paragraph and have modified the section “GcvB inhibits biosynthesis and import of AAAs” in the revised manuscript.

4. Line 210. Inconsistent how?

This sentence was omitted in the revised manuscript.

5. Line 217. How can you say "significant" without statistics?

The growth assay was performed with three independent experiments (n = 3) as described in the legend of Fig. 6. Nonetheless, we have omitted “significant” in this sentence.

6. Table 2. What does the red indicate?

Corrected.

7. Figure 4D. Please include GcvB expression levels between vector and +GcvB in your northern?

We have added the results of northern blot to show that ectopic expression of GcvB is induced by arabinose to similar levels in LB and M9 media.

Re: Spectrum02035-24R1 (Post-transcriptional regulation of aromatic amino acid metabolism by GcvB small RNA in *Escherichia coli*)

Dear Dr. Masatoshi Miyakoshi,

thank you for submitting a revised version.

The reviewers acknowledge your efforts and the revision. However, there are still some minor points that I would ask you to consider in a further revised version before publication.

Revision Guidelines

Sincerely,
Tino Polen
Editor
Microbiology Spectrum

Reviewer #1 (Comments for the Author):

The manuscript can be accepted.

Reviewer #2 (Comments for the Author):

The revised manuscript by Kanda et al. is improved from the first version. However, additional modifications are required prior to publication.

1. You still did not adequately explain the construction of your translational fusions. In the methods you need to specify the sequences of each gene that are included in each fusion. Indicate that the fusion includes from -X to +Y relative to the start of transcription, which is +1. You should also indicate that then fusions contain the gene's promoter, SD sequence, and the beginning of the coding sequence fused in frame with gfp.

2. Line 170. Why is Fig. 2C referenced here?

3. Lines 170-172. If the Tyr pathway genes are not regulated, how do you explain the R1-dependent repression of tyrA in Fig. 3B?

3. Lines 219-226. This paragraph is confusing because you jump back in forth between descriptions of lag time and growth. By growth I assume you mean exponential phase growth rate. Be clear in each instance if you are referring to lag time and growth rate. I think you should include a Figure 6F with a table of growth rates (i.e., doubling times during exponential phase).

4. Table 2. There were several strains used in this study in addition to those listed in the table. Each strain with a plasmid is a new strain. Include all strains in this table.

Reviewer #2 (Comments for the Author):

The revised manuscript by Kanda et al. is improved from the first version. However, additional modifications are required prior to publication.

1. You still did not adequately explain the construction of your translational fusions. In the methods you need to specify the sequences of each gene that are included in each fusion. Indicate that the fusion includes from -X to +Y relative to the start of transcription, which is +1. You should also indicate that then fusions contain the gene's promoter, SD sequence, and the beginning of the coding sequence fused in frame with gfp.

We have provided new supplementary tables 3 and 4 in the revised manuscript.

2. Line 170. Why is Fig. 2C referenced here?

Because aroF-tyrA is a bicistronic mRNA and the aroF mRNA level is shown in Fig. 2C.

3. Lines 170-172. If the Tyr pathway genes are not regulated, how do you explain the R1-dependent repression of tyrA in Fig. 3B?

We think that the translational reporter assay is useful to screen a plethora of sRNA target candidates, but the length of coding sequence may alter the secondary structure of mRNA in some cases. The translational fusion contains the 190-bp aroF-tyrA intergenic region but might not be sufficient to recapitulate the post-transcriptional regulation by GcvB. In a follow-up study, western blot, enzymatic assay, and in vitro assays such as ribosome toeprinting and cell-free translation with PURE system could be performed to clarify whether GcvB regulates the translation of tyrA without affecting the mRNA stability.

3. Lines 219-226. This paragraph is confusing because you jump back in forth between descriptions of lag time and growth. By growth I assume you mean exponential phase growth rate. Be clear in each instance if you are referring to lag time and growth rate. I think you should include a Figure 6F with a table of growth rates (i.e., doubling times during exponential phase).

We have rewritten the paragraph in the revised manuscript.

4. Table 2. There were several strains used in this study in addition to those listed in the table. Each strain with a plasmid is a new strain. Include all strains in this table.

We agree that strains harboring different plasmids should be regarded as different strains, but listing all combinations of sRNA-expression plasmids and translational reporter plasmids is lengthy. Previous papers using the same two-plasmid reporter system do not list all strains but plasmids (e.g. Sharma et al., 2007, 2011).

Re: Spectrum02035-24R2 (Post-transcriptional regulation of aromatic amino acid metabolism by GcvB small RNA in *Escherichia coli*)

Dear Professor Masatoshi Miyakoshi,

your manuscript has been accepted, and I am forwarding it to the ASM production staff for publication. Your paper will first be checked to make sure all elements meet the technical requirements. ASM staff will contact you if anything needs to be revised before copyediting and production can begin. Otherwise, you will be notified when your proofs are ready to be viewed.

Sincerely,
Tino Polen
Editor
Microbiology Spectrum